# EGFAFS: A Novel Feature Selection Algorithm Based on Explosion Gravitation Field Algorithm

**DOI:** 10.3390/e24070873

**Published:** 2022-06-25

**Authors:** Lan Huang, Xuemei Hu, Yan Wang, Yuan Fu

**Affiliations:** 1Key Laboratory of Symbol Computation and Knowledge Engineering of Ministry of Education, College of Computer Science and Technology, Jilin University, Changchun 130012, China; huanglan@jlu.edu.cn (L.H.); huxm18@mails.jlu.edu.cn (X.H.); 2School of Artificial Intelligence, Jilin University, Changchun 130012, China; 3Institute of Biological, Environmental and Rural Sciences, Aberystwyth University, Aberystwyth, Ceredigion SY23 3FL, UK; fu.yuan730@gmail.com

**Keywords:** heuristic algorithm, Explosion Gravitation Field Algorithm, feature selection, gene expression data

## Abstract

Feature selection (FS) is a vital step in data mining and machine learning, especially for analyzing the data in high-dimensional feature space. Gene expression data usually consist of a few samples characterized by high-dimensional feature space. As a result, they are not suitable to be processed by simple methods, such as the filter-based method. In this study, we propose a novel feature selection algorithm based on the Explosion Gravitation Field Algorithm, called EGFAFS. To reduce the dimensions of the feature space to acceptable dimensions, we constructed a recommended feature pool by a series of Random Forests based on the Gini index. Furthermore, by paying more attention to the features in the recommended feature pool, we can find the best subset more efficiently. To verify the performance of EGFAFS for FS, we tested EGFAFS on eight gene expression datasets compared with four heuristic-based FS methods (GA, PSO, SA, and DE) and four other FS methods (Boruta, HSICLasso, DNN-FS, and EGSG). The results show that EGFAFS has better performance for FS on gene expression data in terms of evaluation metrics, having more than the other eight FS algorithms. The genes selected by EGFAGS play an essential role in the differential co-expression network and some biological functions further demonstrate the success of EGFAFS for solving FS problems on gene expression data.

## 1. Introduction

In recent years, a huge amount of data has been produced from several domains, such as business, economics, industry, biology, and medicine. It is difficult to obtain useful information from these raw data, with many points being irrelevant or redundant. A very high-dimensional dataset chokes the models due to the exponentially increased training time and an increased risk of overfitting. Therefore, feature selection (FS) is vital in data mining and machine learning model development for preparing the data properly [1]. The primary objective of FS is to find the optimal subset with representative features from the original feature space by removing irrelevant and redundant features. The gene expression data often consist of a few samples, along with high-dimensional features [2,3,4]. Irrelevant or redundant features lead to unsatisfactory classification accuracy and make it difficult to find potentially meaningful knowledge [5,6,7]. The main target of FS is to find a subset with highly representative features that result in satisfactory classification accuracy. Therefore, choosing an appropriate FS method for gene expression data is essential in obtaining satisfactory classification accuracy.

In recent decades, researchers from various fields have conducted a wide range of studies on aspects of FS, including statistics, data mining, pattern recognition, and bioinformatics data analysis [8,9,10,11]. These widely used FS methods are classified into four categories: (1) Filter-based methods evaluate features depending on their inherent properties, such as the statistical properties of the data. The most popular filter methods are chi-square [12], the gain ratio [13], information gain [14], ReliefF [15] and minimum Redundancy Maximum Relevance (mRMR) [16]. The main advantages of filter methods are that they are not dependent on classifiers and are fast and straightforward in terms of computation. The filter methods are usually used in low-dimensional feature space. A common disadvantage is that they usually ignore the dependence between different features and process the feature individually; (2) Wrapper methods need to employ optimization algorithms for FS, such as Boruta [17], Genetic Algorithm (GA) [18], Particle Swarm Optimization (PSO) [19], and other heuristic-based methods, to find subsets of features. They usually offer better accuracy than filter methods, but they are computationally intensive for high-dimensional datasets because of the adoption of optimization algorithms; (3) Embedded methods such as EGSG [20], HSIC Lasso [21], and DNN-FS [22] have built-in approaches to select the valuable features. The process of FS is usually one part of the classification model. They are less computationally intensive than the wrapper methods, and the link between the classifier and FS is tighter; (4) Hybrid methods are usually the combination of other approaches. The filter and wrapper methods are generally merged to form hybrid methods [23]. Hybrid methods utilize the advantages of the two kinds of methods. They offer better accuracy and computational complexity than filter-based and wrapper methods. All in all, the four categories of FS methods each have their own advantages and disadvantages. 

FS is a common discrete optimization problem. An FS problem with *d* features contains $2^{d}-1$ possible subsets (solutions). The difficulty of FS for high-dimensional data is that the search space increases exponentially with the increment in the size of the features. Then, how to best choose an appropriate approach to search for the optimal subset from the original feature space is a crucial issue. In recent years, heuristic-based approaches have been used successfully to deal with FS problems [24,25,26].

Heuristic-based approaches can usually solve any optimization problems since the search process is a black box, and no domain knowledge or prior knowledge needs to be obtained in advance. These heuristic-based approaches are randomly initialized with a variety of candidate solution(s) in the search space and iteratively refine the solution(s) based on the heuristic functions. Due to the population mechanism, heuristic-based approaches produce a variety of solutions during a search, which makes it suitable for multi-objective FS [27], especially in dealing with the balance between the size of the feature and the classification accuracy. However, heuristic search does not always ensure an optimal solution, and it still has the risk of falling into local optimal solutions, especially when dealing with high-dimensional data such as gene expression data. Tremendous research efforts are being made in the field of FS; related works are introduced in detail in the next section. 

All kinds of FS methods work well on single or multiple datasets, but there is room for improvement, especially when dealing with data with high-dimensional features. As the theorem “no free lunch” [28] states that no optimization method outperforms other optimization methods for all problems, or even the same problem in different instances, the field of FS remains open to the study of the viability of other methods, such as the Explosion Gravitation Field Algorithm (EGFA). EGFA was proposed for continuous problems by our research group in 2019, mimicking the formation process of planets based on SNDM [29]. In this study, we propose a general FS approach based on EGFA, called EGFAFS. The main contributions of this study can be concluded as follows: (1) A novel FS algorithm based on EGFA is proposed, called EGFAFS. (2) EGFA is applied to combinatorial optimization problems for the first time. (3) A recommended feature pool is constructed for initialization to improve the search efficiency. (4) All features in the original feature space are considered during the explosion process to decrease the probability of the algorithm falling into local optima. (5) Experiments are conducted to verify the performance of our EGFAFS.

The organization of this paper is arranged as follows. Section 2 contains three parts: (1) the introduction of related works, (2) the description of the original EGFA, and (3) the improvement and implementation of EGFA for solving the combinatorial optimization problem: feature selection. Section 3 contains details about the experiments, such as a description of the datasets, the evaluation metrics, the comparison results of different FS methods, etc. Section 4 reports the discussion and Section 5 discusses the conclusions.

## 2. Materials and Methods

### 2.1. Related Works

Feature subset selection using exhaustive search is an NP-hard problem. Using heuristic-based methods to solve optimization problems has become increasingly popular due to their ability to efficiently search for a global optimum. Most heuristic-based algorithms are motivated by natural phenomena. For instance, the Simulated Annealing (SA) algorithm [30] is inspired by annealing in metallurgy. Genetic Algorithm (GA) and Differential Evolution (DE) [31] emulate the process of natural evolution. Particle Swarm Optimization (PSO) simulates social behavior. All of the above heuristic search methods have been extensively used in several domains.

However, regardless of the viability of such classical heuristic search algorithms, they cannot provide optimal solutions for all kinds of optimization problems, especially when dealing with high-dimensional optimization problems. Thus, a range of novel optimization algorithms have been proposed in recent years. In 2014, Xing et al. [32] identified a vast number of novel heuristic algorithms (134 in total) and categorized them into four classes: biology-based (99 in total), physics-based (28 in total), chemistry-based (5 in total), and mathematics-based (2 in total) heuristic search algorithms. The efficacy of these heuristic algorithms has been verified in various domains of research. Accordingly, some heuristic algorithms have successfully been utilized for the FS problem. Some heuristic-based approaches for FS are discussed as follows.

There is a tremendous number of recent studies in the literature that utilize heuristic search algorithms for FS. For instance, in [33], The BBO-SVM-RFE proposed by Dheeb Albashish et al. in 2021 is designed to solve FS problems based on Binary Biogeography Optimization (BBO) followed by the application of Support Vector Machine Recursive Feature Elimination (SVM-RFE). In [34], binary variants of the ant lion optimizer (ALO) are proposed to select the optimal feature subset for classification. In [35], Hossam Faris et al. propose an efficient binary Salp Swarm Algorithm (SSA) with a crossover scheme for FS problems. In [36], a Binary Crow Search Algorithm with Time Varying Flight Length (BCSA-TVFL) is applied to feature selection problems in wrapper mode. In [37], a binary moth-flame optimization (B-MFO) is proposed to select effective features from small and large medical datasets. In [38], a novel GSA-based algorithm with evolutionary crossover and mutation operators is proposed to deal with FS tasks. The authors of [39] propose a novel approach to dimensionality reduction by using the Henry gas solubility optimization (HGSO) algorithm for selecting significant features in order to enhance the classification accuracy. In [40], an improved binary particle swarm optimization (IBPSO) algorithm is proposed to solve the FS problem. Reference [41] proposes four different improved versions of the Sine Cosine Algorithm (ImpSCAs) for FS.

Some heuristic-based algorithms have been used extensively for feature selection on gene expression datasets. For instance, reference [42] proposes a novel distributed method consisting of the MR-based Fisher score (mrFScore), MR-based ReliefF (mrRefiefF), and MR-based probabilistic neural network (mrPNN) using a weighted chaotic grey wolf optimization technique (WCGWO). In [43], a hybrid algorithm is proposed using simulated annealing (SA) and the Rao algorithm (RA) for selecting the optimal gene subset and classifying cancer. Reference [44] proposes the Cuckoo search method as a feature selection algorithm, guided by the memory-based mechanism to save the most informative features that are identified by the best solutions. In [45], Lu et al. introduce a novel hybrid FS method that combines Mutual Information Maximization (MIM) and the Adaptive Genetic Algorithm (AGA), named MIMAGA-Selection. In [46], a hybrid method that uses a Genetic Algorithm with Dynamic Parameter setting (GADP) for feature selection on microarray data is proposed. Chuang et al. [47] propose a new hybrid method for gene selection that combines Correlation-based Feature Selection (CFS) and the Taguchi-Genetic Algorithm (TGA). Reference [48] proposes a novel evolutionary method for gene selection on microarray data based on the Genetic Algorithm (GA) and artificial intelligence, named the Intelligent Dynamic Genetic Algorithm (IDGA).

In addition to the methods introduced above, there are still many recent heuristic-based methods for FS in bioinformatics. They all try to solve FS problems with high-dimensional feature space. The significant advantage of heuristic-based approaches is that they do not need any prior knowledge nor any assumption of the feature search space, such as the space being linearly or nonlinearly separable. The fact that a number of datasets in bioinformatics, such as those containing gene expression data, are composed of mutual dependence and interacting features in high-dimensional search space makes heuristic approaches suitable to process these data. In this study, we propose a general feature selection approach based on EGFA, called EGFAFS. A detailed description of EGFAFS will be introduced in Section 2.3.

### 2.2. The Original Explosion Gravitation Field Algorithm (EGFA)

This section provides a full description of the EGFA in two different parts. The inspiration for the EGFA is given in Section 2.2.1, while the procedural steps of the EGFA are outlined in Section 2.2.2.

#### 2.2.1. Inspiration for EGFA

The Explode Gravitation Field Algorithm (EGFA) is a novel nature-inspired heuristic algorithm and was proposed in 2019 by our research team. The basic idea of EGFA is to simulate the formation process of planets based on SNDM and the Big Bang Theory [49]. In EGFA, an individual is abstracted as a dust particle with four attributes: location, group, flag, and mass. There are two kinds of dust particles in EGFA, the center dust particle with the heaviest mass value in the group, and other dust particles, i.e., the surrounding dust particles. Due to the gravitation field, each center dust particle attracts surrounding dust, and all surrounding dust particles move toward the center dust particle. For every dust particle, the location corresponds to a solution to an optimization problem. The value of the group is an integral number. The flag is a Boolean indicating whether it is a center particle. Additionally, the mass value is calculated by a heuristic function.

#### 2.2.2. Procedural Steps of EGFA

The procedure of EGFA is divided into seven main steps, as shown in Figure 1. These steps help other researchers to use or modify EGFA when utilized to solve optimization problems. The pseudo-code of EGFA is given in Algorithm 1. The procedural steps of EGFA in detail are described as follows:

Step 1. Initialize. Initialize n dust particles in the search space randomly. Calculate their mass value by the mass function, which is usually based on the objective function or the benchmark function. The dust population can be formulated as Equation (1).
(1)$$Dust=\left[dust_{1},dust_{2},\cdots ,dust_{n}\right]$$

Step 2. Group. Randomly divide the dust population (composed of n dust particles) into g groups. For every group, sort the dust particles by the value of their mass, and assign the dust particle with the maximum mass as the center dust particle, the attribute flag of which is updated to value 1.
**Algorithm 1.** The pseudo-code of the Explosion Gravitation Field Algorithm.1. **Input**: the size of dust population n, the No. of groups g, the No. of iterations $T_{\max}$2. Initialize the dust population of size n, calculate the mass for each dust particle3. **while**
$t<T_{\max}$
**do**4.   Divide the dust population into g groups5.   Move surrounding dust particles to their center by Equations (2)–(4)6.   Some surrounding dust particles are absorbed by their center7.   Explosion strategy produces some dust particles by Equation (5)8.   t=t+1
9. **end while**10. return the best solution

Step 3. Move and rotate. The surrounding dust particles move toward their center dust particle, which remains stationary. The pace of movement is defined as Equation (2).
(2)$$pace_{i}=w\times dis_{i}$$
(3)$$dis_{i}=center.location-dust_{i}.location$$
where $dis_{i}$ is the difference between the location of $dust_{i}$ and its center, w is a weight for dis, and its default value is 0.068 in [50]. In this step, the location for $dust_{i}$ is updated as Equation (4).
(4)$$dust_{i}.location=dust_{i}.lcation+w_{1}*dis_{i}*\left(1-dust_{i}.flag\right)+w_{2}*rand$$
where $w_{1}>0,w_{2}>0$ are the weights, and the default value of $w_{1}=0.068$ is equal to w; the default value of $w_{2}$ is 0.01 in [50].

Step 4. Absorb. Some surrounding dust particles close to the center dust particle are absorbed by their center. In this process, the size of the dust population will decrease. 

Step 5. Explode. Some new dust particles will be generated by the explosion strategy based on the center dust particle. In this process, the size of the dust population will increase to the original scale. The new dust particle generated in this step is represented by the following formulation:(5)$$dust_{i}=center.location+radius*rand$$
where radius is the radius for the explosion strategy.

Step 6. Check the stopping condition. If the stopping condition is not met, go to step 3; otherwise, stop the loop.

The flow chart of EGFA is given in Figure 1.

The original EGFA is proposed to solve continuous optimization problems, such as the Sphere, or Ackley benchmark problems in low-dimensional search space, having 2, 3, 5, 10, or 20 dimensions. The EGFA has achieved excellent performance in terms of efficiency and accuracy. However, at the same time, EGFA faces challenges when dealing with problems in higher dimensions in terms of accuracy and running time, like all population-based search algorithms.

### 2.3. EGFA for Feature Selection

It is well known that FS is an NP-hard problem. Finding the best subset from a high-dimensional search space is a combinatorial optimization problem. In this study, we utilize EGFA to solve the combinatorial optimization problem, i.e., feature selection for datasets with high-dimensional feature space, called EGFAFS. To verify the performance of EGFAFS, we test EGFAFS on eight gene expression datasets with high-dimensional features. It is noted that the concept of “feature selection” for a general dataset corresponds to the concept of “gene selection” for gene expression data. Since gene expression data usually consist of relatively few samples characterized by high-dimensional features, simple feature selection methods, such as filter-based methods, are not suitable to deal with gene expression data. Our proposed EGFAFS constructs a recommended feature pool for initialization. Attaching more attention to the features in the recommended feature pool, rather than all features in the original feature space, can help find the best solution more efficiently. In addition, considering all features in the original search space during the Explosion process of EGFAFS decreases the probability of being trapped in local optima.

#### 2.3.1. Construct a Recommended Feature Pool

Since gene expression data consist of high-dimensional features, we construct a recommended feature pool based on a series of Random Forests (RF) [51]. Based on the recommended feature pool, we are able to pay more attention to the features in the recommended feature pool, rather than all features in the original feature space. To build the recommended feature pool, we utilize Random Forest (RF) to measure the importance of each input feature based on the Gini coefficient. The pseudo-code of this strategy is given in Algorithm 2.
**Algorithm 2.** The pseudo-code of constructing a recommended feature pool.1. **Input**: The No. of RFs num, the No. of features selected for every RF c, the size of the recommended feature pool q2. **While**
i<num
**do**3.   Sample c features from the original feature space randomly for $RF_{i}$
4.   Feed the samples sliced by c features to $RF_{i}$
5.   Compute the importance scores for features in $RF_{i}$ in Equations (6)–(10)6. **End while**7. Merge all the importance scores for all features of the num RFs8. Rank all features by sorting the scores of importance9. return q features with maximum scores of importance to build the recommended feature pool

The procedural steps of this strategy are described in detail, as follows:

Step 1. Use the Python package named sklearn.ensemble to initialize num Random Forests (RFs).

Step 2. Randomly sample c features from the original feature space (19,214 features) for num times as the input features for the RFs initialized by Step 1.

Step 3. Feed the samples with c features picked by Step 2 to every RF generated by Step 1 for training.

Step 4. Compute the importance scores of features in RF based on the Gini index.

Step 5. Merge all the importance scores of features computed by num RFs.

Step 6. Rank all the features by sorting the scores of importance obtained by Step 5; the q features with maximum scores of importance are composed of the recommended feature pool.

In this study, we set num=2000, c=50, and q=300. Due to the recommended feature pool jointly determined by num=2000 Random Forests, the default values of all parameters for Random Forests are adopted. Specifically, n_estimators=100, criterion=“gini”, min_samples_split=2, and min_sample_leaf=1. The detailed description of Step 4 is as follows.

A Random Forest is composed of several decision trees (binary trees), and the features selected are used to decide to which class the input data belong. In this study, the average loss of the entropy criterion, such as the Gini index, is adopted for growing decision trees. The Gini index for each split node m in each decision tree is presented as $GI_{m}$ and can be calculated using Equation (6).
(6)$$GI_{m}=\sum_{1}^{K}p_{mk}(1-p_{mk}),$$
where K is the number of classes, and $p_{mk}$ is the proportion belonging to the k-th class in node m. Then, the score of the importance of feature $x_{j}$ in node m is presented as $VIM_{jm}^{gini}$, which can be calculated by Equation (7).
(7)$$VIM_{jm}^{gini}=GI_{m}-GI_{l}-GI_{r},$$
where $GI_{l}$ and $GI_{r}$ are the Gini indexes of newly generated nodes after node m splitting. M is defined as the set of nodes that selects the feature $x_{j}$ in the i-th tree. Then, for the i-th tree, the score of importance for feature $x_{j}$ can be calculated by Equation (8).
(8)$$VIM_{ij}^{gini}=\sum_{m\in M}VIM_{jm}^{gini}$$

Therefore, if a Random Forest consists of n trees, the score of importance for feature $x_{j}$ can be calculated using Equation (9).
(9)$$VIM_{ij}^{gini}=\sum_{i=1}^{n}VIM_{ji}^{gini}$$

The final score is given as Equation (10) after normalization.
(10)$$VIM_{j}=\frac{VIM_{j}}{\sum_{i=1}^{c}VIM_{i}}$$

#### 2.3.2. EGFA for Feature Selection Based on a Recommended Feature Pool

The original EGFA is a novel nature-inspired heuristic search algorithm proposed by our research team in 2019 for continuous optimization problems. Detailed information about EGFA is introduced in Section 2.2. This paper utilizes EGFA to solve the combinatorial optimization problem, i.e., feature selection, called EGFAFS. To investigate the performance of EGFAFS, we test EGFAFS on eight gene expression datasets. Because gene expression data consist of high-dimensional features, we construct a recommended feature pool by ranking the features’ importance based on a series of Random Forests. Then, we can pay more attention to the features in the recommended feature pool rather than all features in the original search space. The introduction of this strategy is described in Section 2.3.1 in detail. The overall flow chart of EGFAFS is depicted in Figure 2. The pseudo-code of EGFAFS is given in Algorithm 3.

Step 1. Construct a recommended feature pool based on a series of Random Forests by ranking the scores of features’ importance. This step is described in Section 2.3.1 in detail. After this step, we can obtain the subset M of features with size q, which is the recommended feature pool.

Step 2. With initial *n* dust particles, the location of i-th dust is defined by Equation (11).
(11)$$dust_{i}.location=\left[x_{i},x_{2},\cdots x_{j},\cdots ,x_{c}\right],x_{j}\in M,$$
where $x_{j}$ is the j-th feature selected randomly from the recommended feature pool M, which is built in Step 1. c is the number of selected features. Then, the location of each dust particle is a feature subset with size c. We set n=50, and c=50. In this study, the mass value for each dust particle is calculated based on the Matthews Correlation Coefficient (MCC) [52] as Equation (12).
(12)$$dust_{i}.mass=e^{MCC_{i}},\;$$
where MCC is a metric to evaluate the dust (solution). A detailed description of MCC is given in Section 3.2. In addition, to simplify the process of EGFAFS, the number of groups is set as 1 in this study. There is then only one center in the population. Except for the center dust particle, the other dust particles are the surrounding dust particles.
**Algorithm 3.** The pseudo-code of EGFA for feature selection (EGFAFS).1. **Input**: The size of the recommended feature pool q, the size of dust population n, the size of the features found by EGFAFS c, the No. of Random Forests (RF) num, the No. of iterations $T_{\max}$2. Construct a recommended feature pool by RFs based on the Gini index3. Initialize the dust population of size n by Equation (9), calculate the mass of each particle by Equation (10)4. **while**
$t<T_{\max}$
**do**5.   Move the surrounding dust particles toward their center as Figure 36.   Some surrounding dust particles are absorbed by their center7.   Explosion strategy produces some dust particles as Figure 48.   t=t+1
9. **End while**10. Return the optimal subset of features

Step 3. Move the surrounding dust particles toward their center. For the i-th surrounding dust particle ($dust_{i}.flag=0$), this step is depicted in Figure 3.


Select a feature from the location of $dust_{i}$ randomly, which is named r.Select a feature from the location of its center randomly, which is named s.Replace the feature r by feature s.


In addition, the features represented by the center will be shuffled in this step, which is the strategy of rotating used in this study. 

Step 4. The surrounding dust particles with a relatively smaller value of mass are absorbed by their center. Define basic_mass as the p-th percentile of the mass values for population dust. For $dust_{i}$ (assume that it is surrounding dust, i.e., $dust_{i}.flag=0$), if $dust_{i}.mass<basic\_mass$, $dust_{i}$ will be absorbed; otherwise, it will remain for the next iteration. In this step, the size of the population will decrease. In addition, we set p=80.

Step 5. Several dust particles will be generated by the explosion strategy based on the center dust. The new $dust_{i}$ is generated as shown in Figure 4, which consists of the following procedures.


Copy center.location to $dust_{i}.location$.Select u features from ceter.location randomly as S (consists of red highlighted features), and select u features from the original feature space as O (consists of green highlighted features). We set u=5 in this study.For $dust_{i}.location$, replace the features in S with features in O one by one.


The main loop from Step 3 to Step 5 will run for several epochs (50 epochs in this study). During the main loop, to evaluate a dust particle (a solution, i.e., a subset of features), we feed the training data sliced by the subset of features represented by the dust to the Support Vector Classification (SVC) [53] for training. After training, we utilize the Matthews Correlation Coefficient (MCC), which is a metric for evaluating the performance of classification on the validation dataset, in order to evaluate the dust. A detailed description of MCC is given in Section 3.2. Once the main loop ends, the best subset of features will be found, and the features selected are tested on the independent test dataset to obtain the final performance metrics.

It is noted that the original EGFA was proposed for continuous optimization problems, such as Ackley and Sphere benchmark problems, but that EGFAFS is an improved version of EGFA for solving combinatorial optimization problems (i.e., feature selection) for real-world data, such as gene expression data. Due to the different types of optimization problems solved by EGFAFS and the original EGFA, the processes of Move and Explode in EGFAFS are different from those in the original EGFA, as introduced in Section 2.2. In addition, the original EGFA has good performance for continuous optimization problems in low-dimensional search space, having 2, 3, 5, 10, or 20 dimensions, but the EGFAFS proposed in this manuscript is improved and implemented to solve a combinatorial optimization problem, i.e., feature selection, in high-dimensional search space, containing 19,214 dimensions.

Finally, we have to emphasize that EGFAFS is a general feature selection algorithm which is capable of processing any type of numerical dataset, such as gene expression data.

## 3. Experimental Results and Discussion

### 3.1. Datasets

The Cancer Genome Atlas (TCGA) [54], a landmark cancer genomics program, molecularly characterized over 20,000 primary cancers and matched normal samples spanning 33 cancer types. In this study, eight expression datasets (HNSC, LIHC, LUAD, LUSC, PRAD, READ, STAD, THCA, and UCEC) sliced by 19,214 protein-coding genes are adopted, sourced from TCGA. The eight original datasets consist of more than 60,000 features (expression values), which are marked by Ensembl id [55]. Among these expression values, 19,214 expression values are annotated as protein-coding genes. The eight expression datasets sliced by the 19,214 protein-coding genes are used to verify the performance of EGFAFS and another eight well-known FS methods. Each dataset is composed of approximately 300–600 samples but with 19,214 features (i.e., 19,214 protein-coding genes).

In this study, each dataset we employ has few samples but is characterized by high-dimensional feature space. Additionally, for each dataset, the number of cases of each type of tumor is approximately ten times greater than the number of normal cases, which increases the difficulty of distinguishing between the tumor cases and the normal cases. Detailed information about the eight datasets is given in Table 1.

### 3.2. Evaluation Metrics

To comprehensively evaluate the performance of EGFAFS, seven widely used evaluation metrics are employed, including accuracy (ACC), F-measure (F1), Recall, Precision (PRE), Matthew’s correlation coefficient (MCC), the area under the Precision and Recall curve (AP) and the area under the ROC curve (AUC) [56] to measure EGFAFS against five widely used methods. The metrics are calculated by the following Equations:(13)$$\mathrm{ACC}=\frac{\mathrm{TP}+\mathrm{TN}}{P+N},$$
(14)$$F1=\frac{2\times \mathrm{TP}}{2\times \mathrm{TP}+\mathrm{FP}+\mathrm{FN}},$$
(15)$$\mathrm{Recall}=\frac{\mathrm{TP}}{\mathrm{TP}+\mathrm{FN}},$$
(16)$$\mathrm{PRE}=\frac{\mathrm{TP}}{\mathrm{TP}+\mathrm{FP}},$$
(17)$$\mathrm{MCC}=\frac{\mathrm{TP}\times \mathrm{TN}-\mathrm{FP}\times \mathrm{FN}}{\sqrt{\left(\mathrm{TP}+\mathrm{FP})(\mathrm{TP}+\mathrm{FN})(\mathrm{TN}+\mathrm{FP})(\mathrm{TN}+\mathrm{FN}\right)}},$$
where TP, TN, FP, and FN represent the number of true positives, true negatives, false positives, and false negatives, respectively. In our research, tumor cases are labeled as the positive class and normal cases are labeled as the negative class.

The metrics (ACC, F1, Recall, PRE, and MCC) are defined as Equations (13)–(17). The PR curve is a curve with the Precision (PRE) and the Recall rate as the axes, and the area under the PR curve is the AP. The ROC curve is a curve with the false positive rate (FP-rate) and the true rate (TP-rate) as the axes, and the area under the curve is the AUC. 

The value of ACC, F1, Recall and PRE range from 0 to 1, MCC ranges from −1 to 1, and AP and AUC usually range from 0.5 to 1. In general, the closer these values are to 1, the better the performance they represent.

### 3.3. Verification of the Effectiveness of the Recommended Feature Pool

This section introduces the advantages of using the recommended feature pool. The number of iterations for EGFAFS is set to 50, the number of the Random Forests is set to 2000, the size of the recommended feature pool is set to 300, and the classifier is SVC.

Both the EGFAFS initialized based on the recommended feature pool and the EGFAFS initialized in the original feature space are tested on the eight gene expression datasets. Regarding the three datasets: LUSC, THCA, and UCEC, the former has the same performance as the latter in all metrics (ACC = 1.0000, F1 = 1.0000, Recall = 1.0000, PRE = 1.0000, MCC = 1.0000, AP = 1.0000, AUC = 1.0000). The comparison results for the other five gene expression datasets are shown in Table 2. Part A shows the results of EGFAFS initialized based on the recommended feature pool. Part B shows the results of EGFAFS initialized in the original feature space. The comparison results show that adopting the recommended feature pool can achieve better performance than not using this strategy. In addition, after initialization based on the recommended feature pool, fewer iterations are needed to find a satisfactory subset.

### 3.4. Analysis of the Parameters for EGFAFS

#### 3.4.1. Analysis of the Size of The Target Subsets

The parameter c is the number of features for the subset found. For a thorough investigation, EGFAFS is tested to find the feature subsets with different sizes on eight datasets; more specifically, the size of the feature subsets is set to 30, 50, 100, and 150. The comparison results are shown in the Appendix A. The results show that the size of the target subset has little effect on the accuracy of classification. Specifically, EGFAFS has better performance when the number of the features is set to 30 for some datasets. However, for gene expression data, the correctly classified genes (features) are not necessarily the best pathogenic genes (features). Therefore, it is necessary to slightly expand the scope of selection so that more possible pathogenic genes can be found. Additionally, we can then draw meaningful conclusions in biological analysis. It is noted that the larger the size of the subset, the longer the running time is. In view of this, we set the number of subsets to c=50 for all FS methods.

#### 3.4.2. Analysis of the Population Size of EGFAFS

The parameter n represents the population size of EGFAFS. To investigate the performance of EGFAFS with different population sizes, we test EGFAFS with n=30, 50, 100 on the HNSC dataset and show the results in Figure 5. The horizontal axis represents the number of iterations, and the vertical axis is the scores of the metrics: MCC. Figure 5 shows that EGFAFS has better performance when n=50 or n=100 than when n=30. It is noted that the running time increases with the increase in the population size. In this study, we set n=50 for EGFAFS.

#### 3.4.3. Analysis of the Size of the Recommended Feature Pool

The parameter q is the size of the recommended feature pool. To investigate the performance of EGFAFS initialized based on recommended feature pools of different sizes, we test EGFAFS with q=100, 300, 500 on the HNSC dataset and show the results in the Appendix A. Appendix A shows that EGFAFS has the best performance on three datasets with q=100, has the best performance on seven datasets with q=300, and has the best performance on two datasets with q=500. Therefore, we set q=300 in this study and construct the recommended feature pool for initialization using EGFAFS and four heuristic-based methods (GA, PSO, SA, and ED).

#### 3.4.4. Analysis of the Number of RFs

The parameter num is the number of Random Forests, which we used to construct the recommended feature pool. We test EGFAFS with num=500, 1000, 1500, 2000 on the HNSC datasets, respectively, and show the results in Figure 6. Figure 6 shows that EGFAFS has more stable performance with num=1500 or num=2000 and has the best performance with num=2000. It is noted that constructing the feature pool takes relatively less time over the whole FS process. Then, we set num=2000 in this study to obtain a more stable performance.

### 3.5. Comparison of the Performance of EGFAFS with Eight FS Methods

To verify the performance of EGFAFS, four heuristic-based FS methods (GA, PSO, SA, and DE) and four other FS methods (Boruta, HSIC Lasso, DNN-FS, and EGSG) are adopted for comparison. The four heuristic-based FS methods are implemented based on a Python package called scikit-opt (https://scikit-opt.github.io/, accessed on 12 March 2022) and initialized based on the recommended feature pool. During training, MCC is employed as the heuristic function for four heuristic-based methods to select features. The value of MCC ranges from −1 to 1, where −1 indicates the prediction is completely wrong, and 1 indicates that the prediction is completely correct. For a fair comparison, the number of the population is set to 50 and the maximum iteration is also set to 50 for five heuristic-based methods (GA, PSO, SA, DE, and EGFAFS). For other methods, the default values of all the parameters are adopted. The target size of the features subset is set at 50, and each dataset is split into a learning dataset and an independent dataset with a proportion of 8:2. The learning dataset is further split into a training and a validation dataset by a random seed with a proportion of 6:4. Different random seeds lead to different training and validation datasets. Each method is trained on a training dataset and is validated on the corresponding validation dataset. After each method converges, the final selected features are evaluated on an independent test dataset. Seven commonly used metrics (ACC, F1, Recall, PRE, MCC, AP, and AUC) are employed to compare the performance.

The detailed comparison results of EGFAFS and eight FS methods in seven metrics are represented in the Appendix A. Because the number of cases of each type of tumor (positive class) is approximately ten times greater than the number of normal cases (negative class) for each dataset, some metrics, such as ACC, Recall, and Precision (PRE), cannot evaluate well the real ability of methods to distinguish the negative cases. Then, we select three metrics with great differences relating to the performance of nine FS methods and show the results in Figure 7.

Figure 7 suggests that our EGFAFS has the best performance in three metrics (F1, MCC, and AP) on eight gene expression datasets. In addition, SA, DE and HSICLssso have better performance than the other FS methods. It is noted that all the heuristic-based methods have a similar performance. Such methods are initialized based on the recommended feature pool, which reduces the dimensions of the feature space to acceptable dimensions so as to improve the accuracy of classification. Detailed information on the comparison results can be found in the Appendix A. All in all, our EGFAFS works well for FS and outperforms the eight well-known FS methods in most evaluation metrics on eight gene expression datasets.

### 3.6. Comparison the Running Time of EGFAFS with Eight FS Methods

To compare the time cost, we test EGFAFS and another eight FS methods on eight datasets. The comparison results are shown in Table 3. The total running time of nine methods on eight datasets is shown in the rightmost column of Table 3. The EGSG has the shortest total running time (approximately 235 s), and Boruta has the longest total running time (approximately 1563 s). Our EGFAFS has the third longest running time (approximately 1091 s) during five heuristic-based FS methods and has the fifth longest total running time during the nine FS methods.

### 3.7. Analysis the Distribution of Genes Found by Nine FS Methods

To investigate whether the genes found by nine FS methods have consistency, we analyzed the distribution of the genes and showed the results in the Appendix A. Appendix A shows that the genes found by nine FS methods have less consistency. All the population-based FS methods (GA, SA, PSO, DE, and EGFAFS) are initialized randomly based on the recommended feature pool, and that leads to different solutions (gene subsets). In fact, for a certain population-based FS method such as GA, the solutions it finds are not always fixed; it usually obtains different solutions with similar evaluation scores. Besides the population-based methods, the gene subsets found by other FS methods including EGSG, Boruta, HSICLasso, and DNN-FS have less consistency too, as shown in Appendix A. No gene selected by DNN-FS has been selected by the other eight FS methods.

### 3.8. Analysis of the Degrees of the Selected Genes in the Differential Co-Expression Network

To further verify the significance of our EGFAFS for feature selection, we utilize GEPIA2 [57] for tumor/normal differential expression analysis on eight datasets. GEPIA2 provides customizable functions such as tumor/normal differential expression analysis, profiling according to cancer type or pathological stage, patient survival analysis, similar gene detection, correlation analysis, and dimensionality reduction analysis, and was developed by Zefang Tang, Tianxiang Chen, Chenwei Li, and Boxi Kang of Zhang Lab, Peking University.

By employing the GEPIA2 tool, we select the genes under the conditions of FC≥2 and q<0.01 from the original feature space (19,214 genes) to construct the differential co-expression networks [58,59] by linking two nodes (genes) when their absolute value of Pearson Correlation Coefficient [60] is larger than 0.5. For the eight gene expression datasets, approximately 500–4000 genes with obvious differences between tumor and normal cases are selected. We can then calculate the degree of each node (gene) of the differential co-expression network. The degrees of the genes (features) selected by EGFAFS in differential co-expression networks for eight datasets are represented in Figure 8.

In this study, we present the median values of degrees in the differential co-expression networks with the symbol M_D, refer to the total number of genes selected by EGFAFS with the symbol No_T, and present the number of genes selected by EGFAFS in the co-expression networks with the symbol No_select. Additionally, we refer to the number of genes, the degree of which is greater than M_D, with the symbol No_G. Detailed information of the genes selected by EGFAFS in differential co-expression networks is given in Table 4 and Appendix A. Figure 8 shows the degrees of genes selected by EGFAFS for LIHC dataset in differential co-expression network.

The results depicted in Table 4 and Figure 8 show that most genes selected by EGFAFS on eight gene expression datasets are in the differential co-expression network, and the degrees of the genes selected by EGFAFS are more than the median value of the whole differential co-expression network. In addition, the fact that the genes selected by our EGFAFS play important roles in the differential co-expression network demonstrates the significance of our EGFAFS for feature selection.

### 3.9. GO Enrichment Analysis for Genes Selected by EGFAFS

To help us to understand the biological meaning behind the genes selected by EGFAFS, we utilize the Database for Annotation, Visualization, and Integrated Discovery (DAVID) [61] to perform enrichment analysis. The DAVID provides a comprehensive set of functional annotation tools, which are powered by the comprehensive DAVID Knowledgebase built upon the DAVID Gene concept and pulls together multiple sources of functional annotations.

In general, the Gene Ontology (GO) [62] database describes knowledge of the biological domain concerning three aspects: Biological Process (BP), Cellular Component (CC), and Molecular Function (MF). A gene is usually involved in multiple biological functions with the above aspects. We feed the 50 genes for each dataset selected by EGFAFS into the enrichment analysis tool, DAVID (2021 Update), to discover enriched function-related gene groups. During this process, all the biological functions are selected under the condition of p<0.05. The closer the value is to zero, the more significant the GO term associated with the group of genes is. The results of GO enrichment on eight datasets with three aspects (BP, CC, and MF) are given in Appendix A. Figure 9 shows the results of GO enrichment on LIHC dataset.

In Figure 9, the IDs of biological functions are represented on the x-axis, the counts of enriched function-related genes are represented on the y-axis, and the three kinds of colors present the three top-level biological functions (BP, CC, and MF). For a particular GO term, the more genes are annotated, the more genes work on the corresponding biological function, and the more significant the term is.

For the HNSC, 40 of the 50 genes are annotated to a GO term in the MF aspect. Moreover, four gene groups with approximately 20 genes are annotated to four GO terms in the CC aspect. The GO terms in the BP aspect are not significant. For the LIHC, a gene group with more than 30 selected genes is annotated to a GO term in the MF aspect. Additionally, two gene groups with more than 20 selected genes are annotated to two GO terms in the CC aspect. For the LUAD and LUSC, a gene group with approximately 20 selected genes is annotated to a GO term in the CC aspect. The GO terms in the BP and MD aspects are not significant. For the PRAD and THCA, all GO terms concerning three aspects are not significant, since the counts of genes for GO terms are less. For the STAD, four gene groups with 20–25 selected genes are annotated to four GO terms in the CC aspect, which is significant. For the UCEC, a gene group with more than 40 genes is annotated to a GO term in the MF aspect, and four gene groups with more than 20 genes are annotated to four GO terms in the CC aspect.

We performed enrichment analysis on the gene sets selected by EGFAFS on eight datasets. For six datasets, the genes selected by EGFAFS are enriched in some GO terms with large counts of genes, and the GO terms are significant, belonging to high-level biological functions: CC and MF. For the other two datasets, the enriched GO terms are not significant due to the smaller counts of enriched genes. The overall results shown in Figure 9 demonstrate that most genes selected by EGFAFS play an essential role in some biological functions.

## 4. Discussion

In this study, we propose a novel FS algorithm based on EGFA, called EGFAFS. To investigate the performance of EGFAFS, we tested EGFAFS on eight gene expression datasets compared with eight FS methods.

To implement the EGFAFS, we first utilize a series of Random Forests to measure the importance of the features based on the Gini coefficient. We then select 300 features with maximum scores of importance to build the recommended feature pool. When this process is completed, we initialize the dust population based on the recommended feature pool and calculate the mass of each dust particle based on a metric Matthews Correlation Coefficient (MCC). The main loop for EGFA consists of three steps: (1) Move and Rotate, (2) Absorb, and (3) Explode. We first pay attention to the features in the recommended feature pool (300 features) in the process of Move and Rotate, and then consider all features in the original search space (19,214 features) in the process of Explode. The experiments demonstrate that we should not only pay attention to the features in the recommended feature pool, but also consider all features in the original feature space, meaning that our EGFAFS works well for feature selection in a high-dimensional search space. 

To verify the performance of EGFAFS, we test our EGFAFS on eight gene expression datasets and compare it with four classical heuristic-based FS algorithms (GA, PSO, SA, and DE) and four other FS methods (Boruta, HSIC Lasso, DNN-FS, and EGSG). Seven commonly used metrics (ACC, F1, Recall, PRE, MCC, AP, and AUC) are employed to evaluate the performance of the five methods. The experimental results show that EGFAFS ensures better classification metrics than the other eight FS approaches.

To further verify the significance of our EGFAFS for feature selection, we utilize GEPIA2 for tumor/normal differential expression analysis and construct differential co-expression networks for eight datasets. The results show that most of the genes our EGFAFS selected from eight gene expression datasets are in the differential co-expression network. In addition, the degrees of the selected genes are greater than the median value of the whole differential co-expression network.

To investigate the biological meaning behind the genes selected by EGFAFS, we utilize the DAVID to perform enrichment analysis. For almost all datasets, the genes selected by EGFAFS are enriched for some biological functions, with large counts of genes. The results demonstrate that the genes selected by EGFAGS play an essential role in some biological functions.

In summary, EGFAFS has good performance for feature selection on eight gene expression datasets, compared with eight other well-known FS methods. In addition, the genes selected by our EGFAFS play an important role in the differential co-expression network and in some biological functions. We believe that EGFAFS works well for feature selection.

## 5. Conclusions

In this study, we developed a feature selection algorithm based on EGFA, called EGFAFS. To reduce the dimensions of the feature space to acceptable dimensions, we constructed a recommended feature pool by a series of Random Forests based on the Gini index. We then paid more attention to the features in the recommended feature pool. To verify the performance of EGFAFS for feature selection, we tested EGFAFS on eight gene expression datasets compared with eight well-known FS methods. The results show that among the nine FS algorithms, EGFAFS has the best performance for feature selection on gene expression data in most evaluation metrics. Further analysis of the differential co-expression network and GO enrichment on eight datasets demonstrates that the genes selected by EGFAFS play an essential role in the differential co-expression network and in some biological functions.

## Figures and Tables

**Figure 1 entropy-24-00873-f001:**
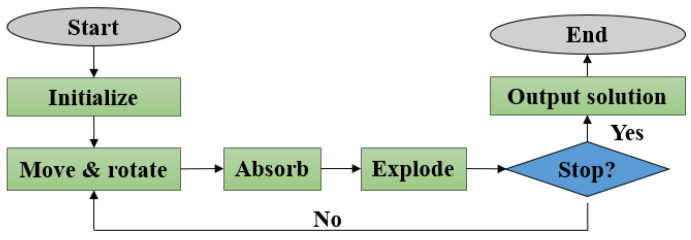
The flow chart of EGFA.

**Figure 2 entropy-24-00873-f002:**
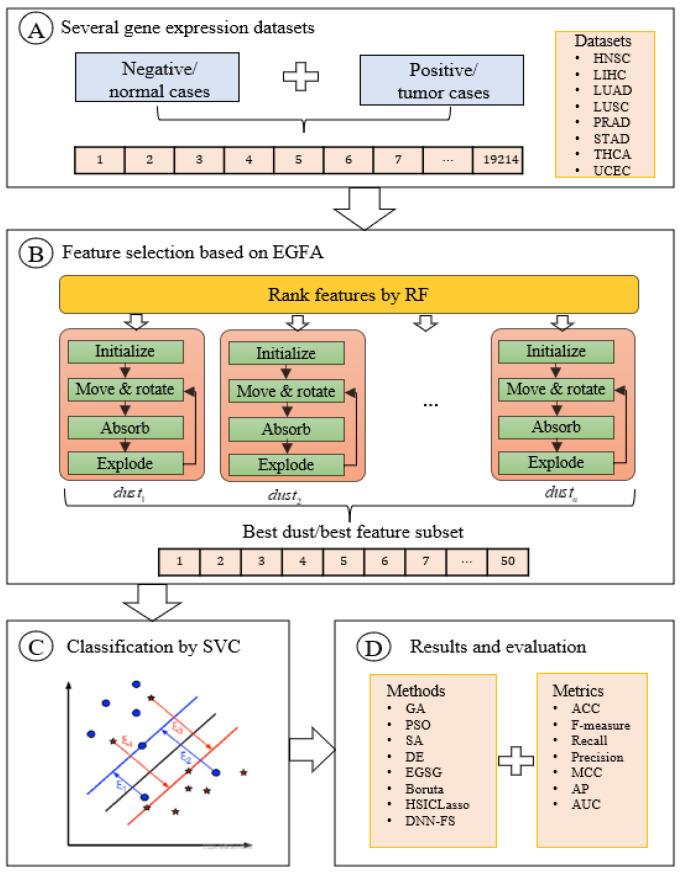
The overall flow chart of EGFAFS.

**Figure 3 entropy-24-00873-f003:**
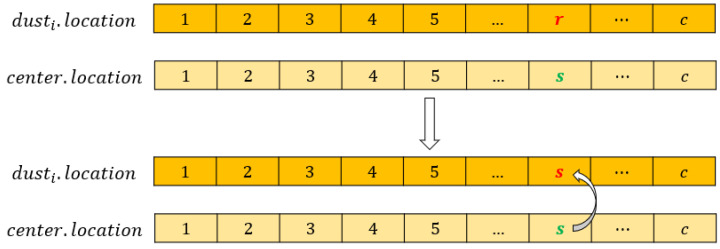
The movement process. r is a feature selected from original $dust_{i}$. s is a feature selected from its center.

**Figure 4 entropy-24-00873-f004:**
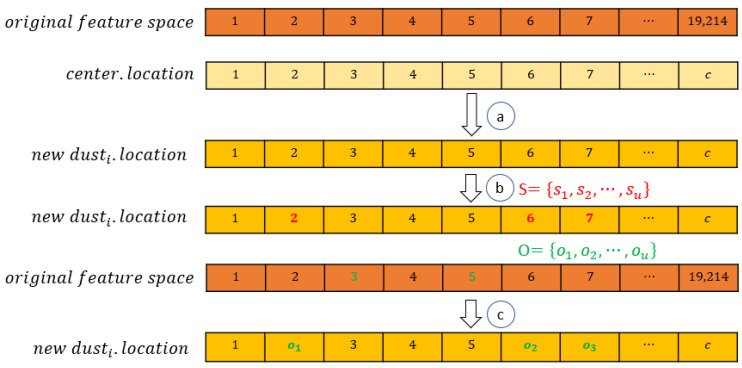
The explosion process. Step a. Copy center.location to $dust_{i}.location$. Step b. Select u features from ceter.location randomly as, and select u features from the original feature space as O. Step c. Replace the features in S with features in O one by one.

**Figure 5 entropy-24-00873-f005:**
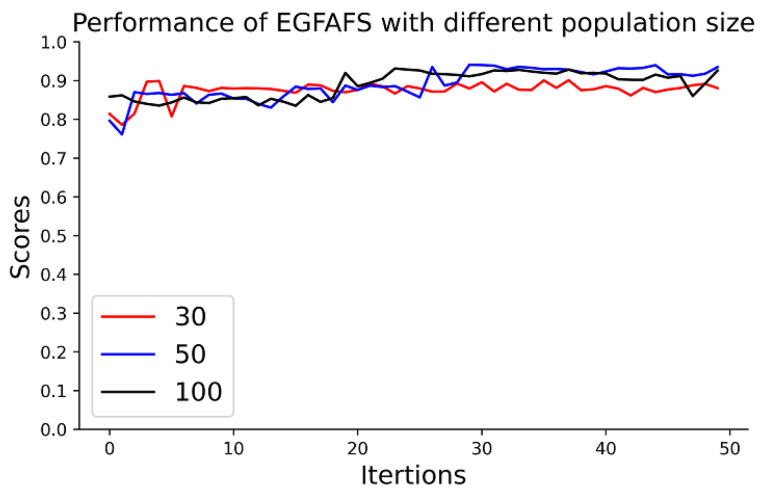
Performance comparison of EGFAFS with different population sizes.

**Figure 6 entropy-24-00873-f006:**
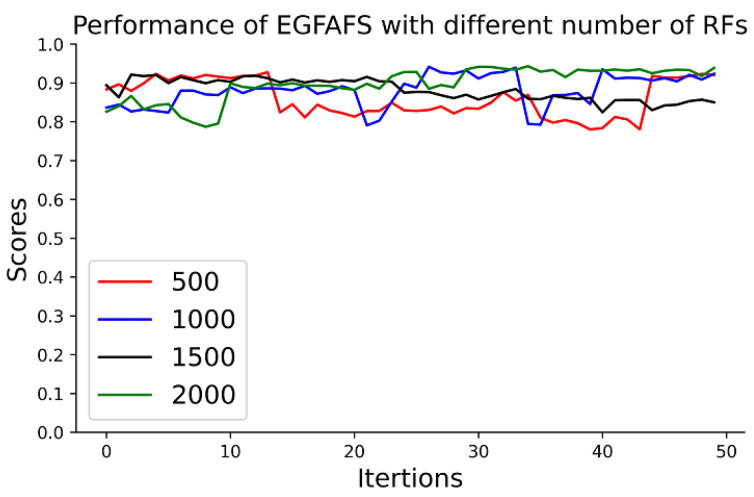
Performance comparison of EGFAFS with different numbers of RFs.

**Figure 7 entropy-24-00873-f007:**
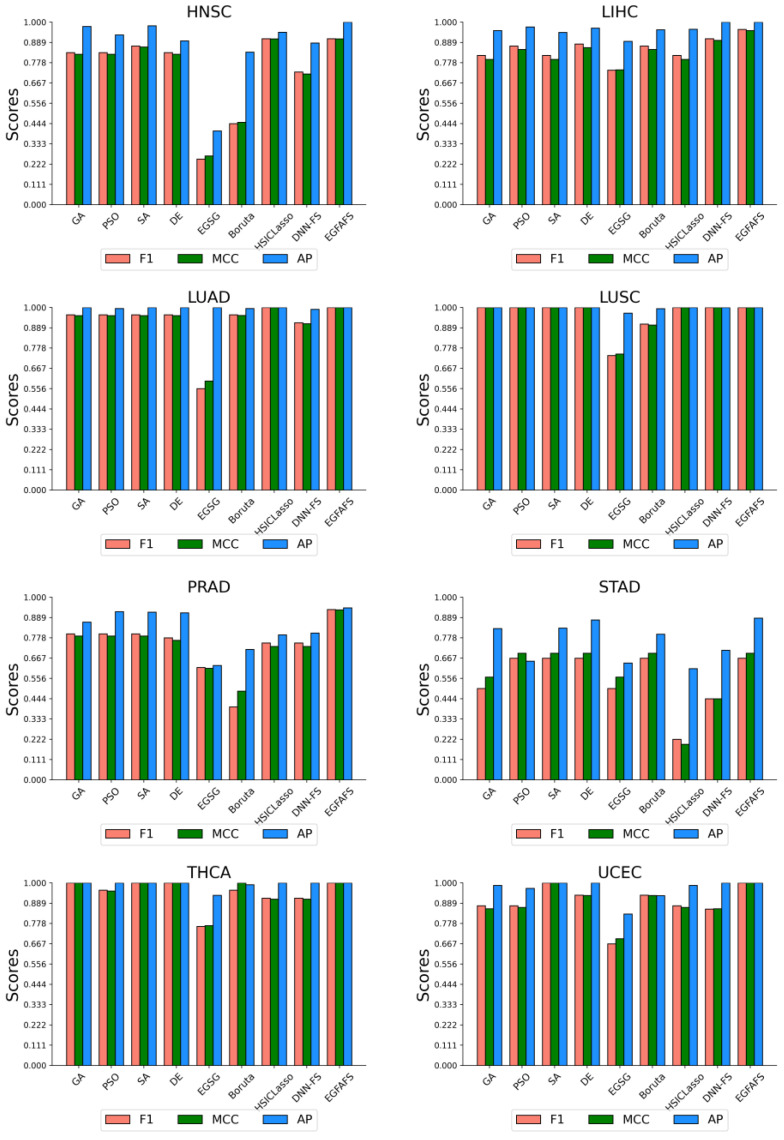
Performance comparison of EGFAFS and eight FS methods on eight datasets. The horizontal axis gives the nine FS methods, and the vertical axis gives the scores of the three metrics: F1, MCC, and AP.

**Figure 8 entropy-24-00873-f008:**
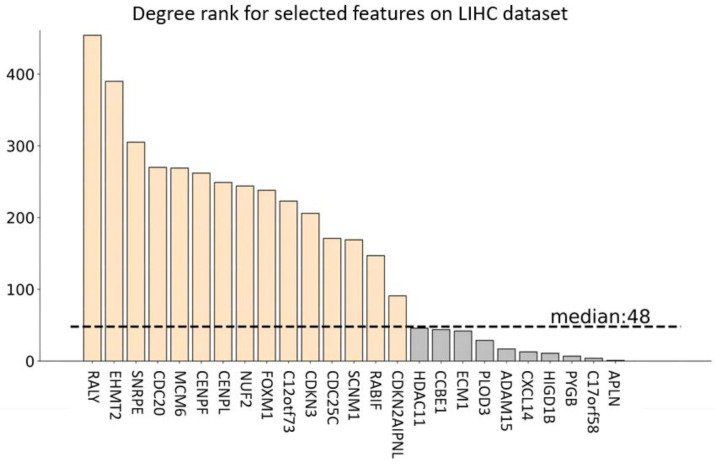
The degree of the features selected by EGFAFS for LIHC dataset in the differential co-expression network.

**Figure 9 entropy-24-00873-f009:**
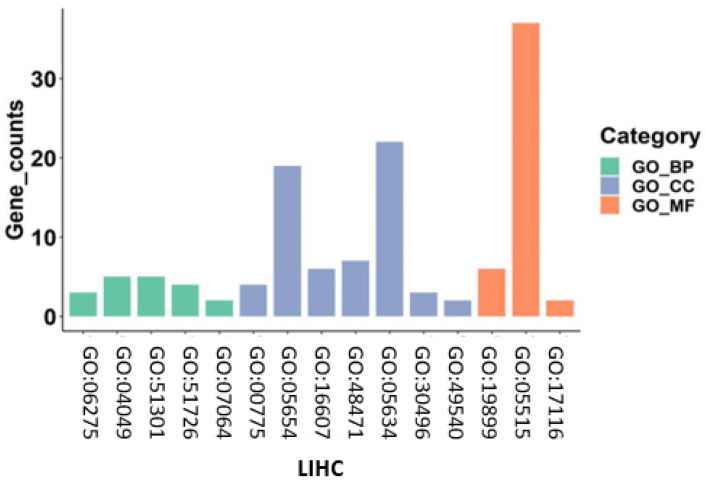
GO enrichment analysis of genes selected by EGFAFS on LIHC dataset.

**Table 1 entropy-24-00873-t001:** Detailed information of eight cancer datasets.

ID	Dataset	No. of Genes	No. of Normal	No. of Tumor
1	HNSC	19,214	44	502
2	LIHC	19,214	50	373
3	LUAD	19,214	59	515
4	LUSC	19,214	49	501
5	PRAD	19,214	52	496
6	STAD	19,214	32	375
7	THCA	19,214	50	510
8	UCEC	19,214	35	544

**Table 2 entropy-24-00873-t002:** The comparison of EGFAFS with and without the recommended feature pool.

	Dataset	ACC	F1	Recall	PRE	MCC	AP	AUC
A	HNSC	**0.9909**	**0.9090**	**0.8333**	**1.0000**	**0.9085**	0.9333	0.9935
	LIHC	0.9764	0.9166	0.9166	0.9166	0.9029	0.9085	0.9897
	LUAD	0.9913	0.9600	0.9230	1.0000	0.9560	0.9897	0.9984
	PRAD	0.9636	0.7142	0.6250	0.8333	0.7035	0.8561	0.9791
	STAD	0.9512	0.6000	0.5000	0.7500	0.5885	0.7164	0.9451
B	HNSC	**0.9909**	**0.9090**	**0.8333**	**1.0000**	**0.9085**	**1.0000**	**1.0000**
	LIHC	**0.9882**	**0.9600**	**1.0000**	**0.9230**	**0.9541**	**1.0000**	**1.0000**
	LUAD	**1.0000**	**1.0000**	**1.0000**	**1.0000**	**1.0000**	**0.9999**	**1.0000**
	PRAD	**0.9909**	**0.9333**	**0.8750**	**1.0000**	**0.9308**	**0.9416**	**0.9914**
	STAD	**0.9634**	**0.6666**	**0.5000**	**1.0000**	**0.6935**	**0.8854**	**0.9817**

**Table 3 entropy-24-00873-t003:** Comparison of the running time of EGFAFS with eight FS methods.

Method	HNSC	LIHC	LUAD	LUSC	PRAD	STAD	THCA	UCEC	Total
GA	167.42	134.33	167.65	155.10	188.16	130.12	172.39	162.00	1277.17
PSO	52.41	43.33	50.38	48.11	59.48	44.53	51.93	51.70	401.87
SA	190.29	148.71	186.36	175.60	221.18	151.78	203.82	187.39	1465.13
DE	79.14	62.51	77.43	70.20	87.37	63.52	79.46	76.00	595.63
EGSG	**29.49**	**26.85**	**30.41**	**30.59**	**30.33**	**26.91**	**30.80**	**29.88**	**235.26**
Boruta	209.31	195.87	191.85	180.71	194.19	197.39	201.10	192.56	1562.98
HSICLasso	48.10	37.22	50.17	52.37	47.23	35.88	47.87	50.32	369.16
DNN-FS	142.47	115.33	148.91	141.82	141.28	109.35	146.92	149.16	1095.24
EGFAFS	143.49	109.79	151.83	138.50	147.37	106.78	153.12	140.49	1091.37

**Table 4 entropy-24-00873-t004:** The information of genes selected by EGFAFS in differential co-expression networks.

ID	Dataset	M_D	No_G	No_Select	No_Total
1	HNSC	30	25	38	50
2	LIHC	48	15	25	50
3	LUAD	56	29	34	50
4	LUSC	52	27	37	50
5	PRAD	108	11	16	50
6	STAD	20	20	29	50
7	THCA	93	16	24	50
8	UCEC	35	45	49	50

## Data Availability

Publicly available datasets were analyzed in this study. Codes and data are available here: https://github.com/abcair/EGFAFS (accessed on 20 May 2022).

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
