# Peer review of "EGFAFS: A Novel Feature Selection Algorithm Based on Explosion Gravitation Field Algorithm"

_entropy, 2022, doi:10.3390/e24070873_

Round 1
Reviewer 1 Report
Review of the manuscript: "EGFAFS: A Novel Feature Selection Algorithm for Gene Expression Data Based on Explosion Gravitation Field Algorithm"
by L. Huang et al.
The manuscript presents an algorithm to properly classify genes (features) based on the corresponsding RNA expression profile (data). The algorithm selects a pool of genes by a series of random forests based on the Gini index. The algorithm is tested on messenger expression data downloaded from TCGA and consisting of eight tumors (samples) and 19214 genes (features). The aim is to fild a subset of genes able to distinguish between the normal and the tumor case. A number of alternarive methods are used and the manuscript shows that the proposed algorithm provides better perfomances.
The manuscript is certainly of interest but the text in not very clear and should be improved. for example the title refers to "gene selection" while in the text this is called "feature selection". In my opinion it is important the the title and the text to be consistent: i.e. remove from the title "gene expression data" or explain in the text that the method works only for gene expression data. The text is very unclear in its present form.
Author Response
We greatly appreciate the advice on the previous version of the manuscript. The following is our point-by-point response to each of the comments or questions. I would like to take this opportunity to thank you again for handling our manuscript.

Reviewer 2 Report
This paper proposes a FS strategy for gene expression data based on the EGFA algorithm.
The authors can find below my concerns.
- Major concerns
1) The paper lacks any novelty: basically, EGFAFS is EGFA with an initial feature set determined via random forests (RFs).
2) In light of point #1, the advantages of using vs. not-using the initialization via RFs should be addressed in the manuscript, otherwise we cannot quantify pros and cons of using such RFs
3) In the Introduction, the authors claim that SVMs and K-NN are wrapper-based methods for feature selection: this is not true. SVMs and K-NN are plain classifiers
4) Counterintuitively, the authors in the Introduction claim that wrapper-methods are computationally expensive and likely to be trapped in local minima. This is correct ... but EFGAFS is also a wrapper-method! In short, EGFAFS is a population-based method such as a genetic algorithm or a differential evolution or a particle swarm optimization: the only difference is that EGFAFS employs the RF-based initialization in order to optimize over a reduced feature pool. For the sake of transparency, the same can also be done with genetic algorithms, differential evolution or particle swarm optimization by either employing the same RF-based initialization of the top of those algorithms or by simply defining a constrained optimization problem where at most *q* features can be selected.
5) The choice of the datasets has to be improved: we have 8 datasets having the *same* number of genes. For a thorough investigation, it's necessary to see your algorithm dealing with different datasets, with different number of records and/or features.
6) How the hyperparameters of the RFs have been tuned?
7) The comparison with different feature selection algorithm needs to be carefully discussed. For example, there is no mention about the number of iterations that population-methods need to go through before returning those solutions: convergence speed is an important aspect of optimization algorithms.
8) There is no sensitivity analysis about the parameters of EGFAFS: the authors set c=50, n=50, q=300, num=2000. How did you choose those values? How the convergence speed changes as a function of those parameters?
9) How do the three weights w, w1 and w2 have to be chosen in Eqs. 2 and 4?
10) Do the genes selected by EGFAFS have also been selected by other FS strategies? It would be interesting to investigate whether there is some sort of "agreement" between different FS strategies. If not, it would also be worth investigating why other FS strategies fail in finding meaningful genes.
- Minor concerns
1) The paper needs a strong re-writing stage. Many sentences are badly contructed or incomplete. Especially in the Introduction, several concepts are repeated many times (e.g., the fact that FS can improve the classification accuracy and the fact that gene expression data are high-dimensional)
2) The last sentence of the Abstract needs to be better framed. Indeed, as it is, it seems that the significance of EGFAFS for FS is demonstrated by means of gene roles in co-expression networks. This is not entirely true: it may be true only in the context of gene expression data, but it's not true in general (or, at least, we don't know if it's true since computational experiment only regard genomic data).
3) The authors claim (page 2, lines 70-71) that a d-dimensional FS problem yields 2^d possible solutions. This is not entirely true: the number of solutions is (2^d)-1 since the combination with all 0's is not a valid solution (i.e., a solution with all 0's means that no features are selected)
4) It is customary to end the Introduction with the paper roadmap (i.e., describing what each section contains)
5) There are two sections numbered 2.3.1
Author Response
We greatly appreciate the advice on the previous version of the manuscript. The following is our point-by-point response to each of the comments or questions. I would like to take this opportunity to thank you again for handling our manuscript. Please see the attachment

Round 2
Reviewer 1 Report
none
Reviewer 2 Report
The authors did a very good job in revising the manuscript.